# DFT Method Used for Prediction of Molecular and Electronic Structures of Mn(VI) Macrocyclic Complexes with Porhyrazine/Phthalocyanine and Two Oxo Ligands

**DOI:** 10.3390/ma16062394

**Published:** 2023-03-16

**Authors:** Denis V. Chachkov, Oleg V. Mikhailov

**Affiliations:** 1Kazan Department of Joint Supercomputer Center of Russian Academy of Sciences—Branch of Federal Scientific Center “Scientific Research Institute for System Analysis of the RAS”, Lobachevskii Street 2/31, 420111 Kazan, Russia; 2Department of Analytical Chemistry, Certification and Quality Management, Kazan National Research Technological University, K. Marx Street 68, 420015 Kazan, Russia; olegmkhlv@gmail.com

**Keywords:** porphyrazine, phthalocyanine, oxo ligand, heteroligand manganese complex, DFT method

## Abstract

By using the data of the DFT quantum chemical calculation in the OPBE/TZVP and B3PW91/TZVP levels, the possibility of the existence of a manganese(VI) heteroligand complex containing porphyrazine or its tetra[benzo] derivative (phthalocyanine) and two oxygen (O^2−^) ligands, which is still unknown for this element, is shown. The parameters of the molecular structure, multiplicity of the ground state, NBO analysis data and standard thermodynamic parameters (enthalpy Δ*H*^0^*_f_*, entropy *S*^0^*_f_* and Gibbs’s energy Δ*G*^0^*_f_* of formation) of each of these metal macrocyclic compounds are presented. Additionally, it is noted that, based on the totality of structural data obtained by the above versions of the DFT method, the existence of a similar complex of manganese with di[benzo] derivative of porhyrazine and two oxygen (O^2−^) ligands seems doubtful.

## 1. Introduction

In studies [1,2,3,4,5,6], a quantum-chemical calculation of the parameters of molecular structures of coordination compounds having [M(**P)**(O)_2_] (**I**), [M(**dbP)**(O)_2_] (**II**) and [M(**Pc**)(O)_2_] (**III**) formulas [where M is Cr, Fe, Co, Ni, and Cu and **P**^2−^, **dbP**^2−^ and **Pc**^2−^ are double deprotonated forms of porphyrazine H_2_**P**, trans-di[benzo]porphyrazine H_2_**dbP** and phthalocyanine H_2_**Pc**, respectively] (Figure 1) was carried out. 

According to the data from these studies, obtained using two versions of the DFT method, namely DFT B3PW91/TZVP and DFT OPBE/TZVP, the possibility of the existence of all three types of complexes I–III, in the case of M=Cr, Fe and Co [1,2,3,4,5], but only of the same type, namely II in the case of M=Cu [6], was shown. In this regard, it should be noted that there are contradictory results for M=Cu: according to the DFT OPBE/TZVP data, complexes I and III can also exist, whereas, according to the DFT B3PW91/TZVP data, they cannot. At the same time, for Cr and Co, the oxidation state in each of these compounds is VI (at least formally), which is the maximum among the reliably established values of this parameter for these chemical elements. In this connection, it seems interesting to consider whether such complexes can be formed in the case of M=Mn, where the oxidation state VI is intermediate between the minimum (–III) and maximum (VII) reliably established for the given 3*d* element. What is remarkable is that any information about manganese compounds I, II and III in the literature devoted to macrocyclic ligands, such as porphyrins, porphyrazines and their various substituted compounds [7,8,9,10,11,12,13,14], and also anywhere else, is absent. Nevertheless, these objects are of enough considerable interest for preparative coordination chemistry and the chemistry of macrocyclic compounds. In addition, they may be helpful from a purely practical point of view, since in principle they might be employed at least as potential catalysts for diverse reactions of inorganic and organic synthesis. In this connection, our study was devoted to establishing the possibility of the existence of complexes having I, II and III formulas using modern methods of quantum-chemical calculation and the density functional theory (DFT) and, in the case of a positive answer to this question, the determination of the quantitative characteristics of their molecular and electronic structures.

## 2. Method

As in our studies [1,2,3,4,5,6] cited above, and also in the earlier ones [15,16,17], the DFT B3PW91/TZVP with a combination of B3PW91 functional [18,19] and TZVP basis set [20,21] was used in this study. According to data [22], this DFT version has a minimum of “normal error” compared with other DFT versions. Such conclusion was confirmed with a comparison of the results of determination of the parameters of molecular structure for 3*d*-element macrocyclic complexes with phthalocyanine, obtained using different DFT versions, with the experimental values of these parameters. Additionally, for comparison as in our studies [23,24,25], quantum-chemical calculations were done using the DFT OPBE/TZVP version combining TZVP and the OPBE functional [26,27]. For the 3d-element coordination compounds, the indicated DFT version more adequately predicted the relative energy stabilities of high-spin and low-spin states and also reliably described key geometric parameters of molecular structures of these compounds [27,28,29,30,31]. Calculations were performed using Gaussian09 software [32]. The data obtained as a result of calculations were visualized using the *ChemCraft* 1.8 program. The correspondence of the discovered stationary points to energy minima was proved in all cases using the calculation of energy-second derivatives with respect to atom coordinates, whereinto all equilibrium structures corresponding to minima of the potential energy surfaces had only real positive frequency values. In accordance with the theory of the structure of the atoms, Mn(VI), which is in the complexes I–III, must have 3*d*^1^ electronic configuration, and that is why spin multiplicities (*M_S_*) 2 and 4 were considered for the central ion indicated above. Among the structures optimized at such multiplicities, the one with the lowest total energy was selected. To calculate the parameters of molecular structures with multiplicity greater than 1, we used the unrestricted method (*UB3PW91* and *UOPBE*). In addition, the energetically most favorable structure was always checked according to the STABLE = OPT procedure, while what was important, the wave function corresponding to this structure, was stable in all cases. Natural Bond Orbital (NBO) analysis was carried out using NBO 3.1 version in the framework of Gaussian09 software [32], according to the methodology submitted by [33]. The standard thermodynamic parameters of a formation (Δ*H*^0^*_f_*, *S*^0^*_f_* and Δ*G*^0^*_f_*) for the metal macrocyclic compounds indicated above were computed employing the methodology [34].

## 3. Results and Discussion

According to the data obtained as a result of our quantum-chemical calculation carried out using the B3PW91/TZVP as well as OPBE/TZVP methods, one can state with sufficient certainty the existence of two of the three complexes of types I–III indicated above, where M=Mn, namely [Mn(**P**)(O)_2_] and [Mn(**Pc**)(O)_2_]. In the case of complex II, the result turned out to be ambiguous, since according to the data obtained with the first of these two DFT methods, a complex with such a structure can exist as an isolated molecule, while according to the data obtained with the second method, it cannot. (In this regard, it is interesting that in the case of complexes similar in composition, where M=Cu, only complex II is capable of existing as an isolated molecule, while complexes I and III, according to the results of calculations with the DFT B3PW91/TZVP method, are not capable [6]). Taking into account this important fact, we will discuss further only the [Mn(**P**)(O)_2_] and [Mn(**Pc**)(O)_2_] complexes, since their existence is predicted using both of these DFT methods.

The chemical bond lengths between atoms and bond angles for complexes I and III, calculated using each of the DFT methods indicated above, are presented in Table 1. The images of the molecular structure of the macrocyclic compounds under examination, obtained with the DFT B3PW91/TZVP method, are shown in Figure 1. In this regard, we would like to note that the images of the structures of each of these two complexes obtained with the OPBE/TZVP method are very similar to images calculated with the B3PW91/TZVP method (see Appendix A, Appendix A). As may be seen from the presented data, the key bond lengths, namely (Mn1N1), (Mn1N2), (Mn1N3) and (Mn1N4) in the [Mn(**Pc**)(O)_2_] macrocyclic complex, are equal to each other in the framework of the DFT B3PW91/TZVP method and within the DFT OPBE/TZVP method; however, in the [Mn(**P**)(O)_2_] macrocyclic complex, they are equal to each other only in pairs. This difference can be explained by the fact that, as is easy to see even with a cursory glance at the structural formula [Mn(**P**)(O)_2_], the nitrogen atoms in the MnN4 chelate node are not completely equivalent to each other, because two of them are bonded to neighboring atom carbons only with single bonds, while the other two have one single and one double bond. However, a very noticeable (more than 10 pm) difference in the lengths of the bonds (Mn1O1) and (Mn1O2), which takes place in the same complex according to the DFT B3PW91/TZVP data, while according to the DFT OPBE/TZVP data, the lengths of these bonds are almost the same, attracts attention (Table 1). In this regard, it should be noted that a similar inequality was noted earlier in other complexes of 3*d* elements formed by porphyrazine and two axial oxo ligands [1,5]. The reason for the inequality between the indicated bond lengths remains unclear. It is interesting that such an inequality is not observed if fluoride anions act as axial ligands [35,36].

The MnN4 chelate node, in the case of [Mn(**P**)(O)_2_], according to the data of both DFT methods used in this article, is almost flat (its deviation from coplanarity does not exceed 0.5°), and in the case of [Mn(**Pc**)(O)_2_], it is perfectly flat. At the same time, characteristically, all the angles in this chelate node still differ from 90° in the first of these complexes, although slightly, and then in the second, they are equal to 90° (Table 1). All four 6-membered metal chelate rings as well as all four 5-membered non-chelate rings that are adjacent to 6-membered metal chelate rings contain one nitrogen atom.

Carbon atoms also have either a perfectly flat structure or practically no different from it. Such a conclusion can be made if we take into account the sums of bond angles in each of these structural fragments, which are either equal to 360.0°, 720.0° and 540.0° or very close to these values. In addition, as can be seen from Table 1, 6-membered chelate rings as well as 5-membered non-chelate rings are almost identical to each other in terms of bond lengths between the corresponding atoms and in terms of the assortment of bond angles contained in them. According to the DFT B3PW91/TZVP method, the oxygen and manganese atoms in both [Mn(**P**)(O)_2_] and [Mn(**Pc**)(O)_2_] complexes are under consideration.

When forming an angle (O1Mn1O2) of 180.0° between them, according to the DFT OPBE/TZVP method, the specified value of the angle (O1Mn1O2) takes place only in the case of [Mn(**Pc**)(O)_2_], while for [Mn(**P**)(O)_2_] it is more than 10° less (Table 1). As for the bond angles (OMnN) formed by the “axial” oxygen atoms, the Mn atom and the nitrogen atoms entering the MnN4 chelate node, in the case of [Mn(**Pc**)(O)_2_], they are equal to each other, and each of them is equal to 90°, while in the case of [Mn(**P**)(O)_2_], they are equal only in pairs, four of them are less than 90°, and four are more than 90° (Table 1). Despite this, as well as the above differences between the structural data obtained with these two independent versions of the DFT method, we can still state that, on the whole, these data are in fairly good agreement with each other. Concluding the discussion of the structural data, we noted that the electric moments of dipole (dipole moments) of the [Mn(**P**)(O)_2_] and [Mn(**Pc**)(O)_2_] complexes studied by us, according to the results of calculations using the DFT B3PW91/TZVP method, are 0.18 and 0.00 Debye units, respectively, and according to the DFT OPBE/TZVP method, they are 0.28 and 0.03 Debye units, respectively. As can be seen from these values, they are quite small, which is in rather good agreement with the key parameters of their structures presented in Table 1.

The ground state of both complexes considered by us in the framework of the DFT OPBE/TZVP method is a spin doublet. According to this method, the nearest excited state (spin quartet) is significantly higher in energy than the ground state (by 140.4 kJ/mol in the case of [Mn(**P**)(O)_2_] and by 96.5 kJ/mol in the case of [Mn(**Pc**)(O)_2_]). Such a situation seems quite natural for tetragonal (pseudo-octahedral) complexes with the 3d1 electron configuration and the coordination number of the central metal ion equal to 6, to which [Mn(**P**)(O)_2_] and [Mn(**Pc**)(O)_2_] belong. The B3PW91/TZVP method gives somewhat different results; according to it, the ground state of [Mn(**Pc**)(O)_2_] is also a spin doublet (the nearest excited state lies higher by 47.4 kJ/mol), while the ground state of [Mn(**P**)(O)_2_] is a spin quartet (although the energy distance between it and the nearest doublet state is only 5.2 kJ/mol). However, according to numerous statistical data for complexes with a configuration of d1, which have a coordination number of 6 and a symmetry group of *D_4h_*, *C_2h_* or *C_2v_*, in our opinion, the ground state is precisely the spin doublet, because the result obtained with the OPBE/TZVP method seems in our case to be more reliable. Images of higher occupied (HOMO) and lower unoccupied (LUMO) molecular orbitals of the studied Mn(VI) macrocyclic compounds are shown in Figure 2.

The key data of NBO analysis for complexes I and III are presented in Table 2, and full data can be found in the Appendix A. It is noteworthy that the values of the effective charge on the Mn atoms in both these macrocyclic compounds, according to the NBO analysis, both in the framework of the DFT B3PW91/TZVP method and in the framework of the DFT OPBE/TZVP method, are much less than +1.00. Furthermore, the effective charges on the oxygen atoms are very different from the values (−2.00). The given fact, as well as the values of effective charges on other atoms present in these complexes (in particular, on donor nitrogen atoms N1, N2, N3 and N4), are a direct indication of a very high degree of electron density delocalization in these metal macrocyclic compounds. The distribution patterns of the spin density in the [Mn(P)(O)_2_] and [Mn(Pc)(O)_2_] complexes are shown in Figure 3.

The standard thermodynamic parameters of formation in the framework of the isobaric process for the given macrocyclic metal chelate under examination (Δ*H*^0^*_f_*, *S*^0^*_f_* and Δ*G*^0^*_f_*) are presented in Table 3. As can be seen from the data of this Table, all values of Gibbs’ energy Δ*G*^0^*_f_* are positive. It follows that neither of these two complexes can be obtained from simple substances formed by the chemical elements that make up these complexes (i.e., C, N, O and Mn). Nevertheless, judging by the data of our DFT quantum chemical calculation, the molecular structure of [Mn(P)(O)_2_] and [Mn(Pc)(O)_2_] and the full set of their geometric parameters can be realized as a whole, and therefore both of them are able to exist (at least in the gas phase in the form of separate isolated molecules).

## 4. Conclusions

According to data obtained as a result of our quantum-chemical calculation, both DFT versions indicated above, namely OPBE/TZVP and B3PW91/TZVP, unambiguously predict the possibility of the formation of heteroligand macrocyclic complexes [Mn(P)(O)_2_] (I) and [Mn(Pc)(O)_2_] (III) with doubly deprotonated forms of porphyrazine and its tetra[benzo] derivative (phthalocyanine); however, on the whole, they do not unambiguously confirm the possibility of the existence of an analogous complex with a doubly deprotonated form of trans-di[benzo]porphyrazine (II). In this connection, attention is drawn to the fact that, in the case of M=Cu, the reverse situation takes place: both of these methods confirm the existence of a complex of a similar type with trans-di[benzo]porphyrazine but do not unequivocally confirm the possibility of the formation of complexes with porphyrazine and phthalocyanine. With this in mind, it can be assumed that the relative stability of type II complexes in the Cr–Cu series increases, while the relative stability of complexes I and III decreases.

In the studied complexes, Mn is bonded to four nitrogen atoms and two oxygen atoms. In accordance with the generally accepted definition of the term “oxidation degree”, which was given in Refs. [1,6], we can rightly assume that the oxidation state of the central atom, namely Mn, in each of the complexes [Mn(P)(O)_2_] and [Mn(Pc)(O)_2_] is +6. The point is that in both of these complexes, the Mn atom forms six bonds by the exchange mechanism with atoms that have a higher electronegativity compared with its own, namely two double Mn=O bonds with two oxygen atoms and two single Mn–N bonds. Since the oxidation state of any chemical element is defined as the modulus of the oxidation degree and is displayed by the corresponding Roman numeral, the oxidation state Mn is exactly equal to VI in both [Mn(P)(O)_2_] and [Mn(Pc)(O)_2_]. The fact that, according to the NBO analysis data, the real charge on the manganese atom in each of these complexes differs significantly from the value of +6.00 Å, is not significant in determining the degree (state) of oxidation, since this parameter is in no way related to the above definition. Be that as it may, in connection with the foregoing, it should now confirm the possibility of the formation of these two compounds in a real experiment.

## Data Availability

No unpublished data were created or analyzed in this article.

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
