# Peer review of "DFT Method Used for Prediction of Molecular and Electronic Structures of Mn(VI) Macrocyclic Complexes with Porhyrazine/Phthalocyanine and Two Oxo Ligands"

_materials, 2023, doi:10.3390/ma16062394_

Round 1

Reviewer 1 Report

Minor Revisions

·         In Methods section, line 76, the authors claim that they used unrestricted functionals in DFT calculations for multiplicity greater than 2. Considering Mn (VI) complexes with multiplicity 2 have an unpaired electron, why did they not use unrestricted functionals for these complexes as well? ·         The atomic distances in table 1 are reported in pm. This parameter is usually reported in nm or Angstrom ·         In line 229 when describing the bonds of the manganese atom the authors wrote that it has two bonds to nitrogen atoms and four to oxygen atoms. In the studied complexes Mn is bonded to four nitrogen atoms a two oxygen atoms.

Author Response

·      In  Methods  section,  line  76,  the  authors  claim  that  they  used  unrestricted  functionals  in  DFT calculations  for  multiplicity  greater  than  2.  Considering  Mn  (VI)  complexes  with  multiplicity  2  have  an unpaired electron, why did they not use unrestricted functionals for these complexes as well?
In  fact,  To  calculate  parameters  of  molecular  structures  with  multiplicity  greater  than  1,  we  used  the unrestricted (UB3PW91, UOPBE) method. This phrase is included in the revised text of the paper (note that a similar remark was made by Reviewer 2).

·      The  atomic  distances  in  table  1  are  reported  in  pm.  This  parameter  is  usually  reported  in  nm  or Angstrom.
In principle, we agree with this remark of our esteemed Reviewer 1, however, we would like to note that in all our articles on quantum chemical calculations and published both in Materials and in other journals of the MDPI publishing house (as well as journals of other publishers - Elsevier, Taylor & Francis, etc.), we used  “pm”  as  the  unit  of  interatomic  distances.  The  units  indicated  by  the  Reviewer,  despite  their prevalence, seem to us not quite suitable: the angstrom, as is known, is an off-system unit, and “nm”, although there is a unit of the SI system, however, if it were used, all interatomic distances presented in Table 1 would have to be expressed as numbers starting with zero and four decimal places (or dots), for example, instead of “109.6 pm” write “0.1096 nm”. This, in our opinion, is inconvenient. In this regard, we would very much like to ask our esteemed Reviewer 1 to preserve the unit of measurement of interatomic distances “pm” 
used in this paper.

·      In line 229 when describing the bonds of the manganese atom the authors wrote that it has two bonds to nitrogen atoms and four to oxygen atoms. In the studied complexes Mn is bonded to four nitrogen atoms and two oxygen atoms.
We agree with this remark of our esteemed Reviewer 1, the second phrase indicated by him is included in the revised text of the paper.

All changes and / or additions to the text of the paper, made by us in accordance with the comments of Reviewer 1, are highlighted in the text with a yellow fill. One of these remarks, which coincides in substance with the similar remark of Reviewer 2, is highlighted in green.

On behalf of authors,
Dr. D.V. Chachkov

Reviewer 2 Report

Reviewing of the article titled "DFT Method Using for Prediction of Molecular and Electronic Structures of Mn(VI) Macrocyclic Complexes with Porhyrazine / Phthalocyanine and Two Oxo Ligands" authored by Denis V Chachkov and Oleg V. Mikhailov.

This paper presents a theoretical study comparing two different functionals on a series of hypothetical manganese complexes. The methodology employed by the authors is known to be reliable for metal complexes in general (B3PW91) and especially for spin states energies with the OPBE functional. The chosen basis set is large and reliable. The study details the optimized geometry, frontier molecular orbitals and charges for the compounds that converged. These results are interesting and deserve to be published. However, I think that the journal Materials is not suitable for the study as it is currently presented because the main property discussed here is the feasibility while we would have liked a more important development on other properties such as magnetism or optical characteristics. 

Also, I find that the display of the results is a bit inverted. For me, the most important result is the electronic structure. The geometry is a consequence of the electronic structure. The differences in distances (MnO) and angles (OMnO) probably are the results of a different electronic state.

I also find that the discussion on the redox state is dismissed too quickly. Yes, the definition is based on formal charges based on a simplistic structural formula does not preclude the use of the resources now provided by quantum chemical calculations. Thus the analysis of natural charges (in SI) show a number of 3d electrons quite different to the imagined 3d1 configuration. The metal ligand sigma and pi interactions seem to be interesting. The FMO representation does not always correspond to the unpaired electron(s). Like for [Mn(P)(O)2] where the LUMO do correspond to empty an t2g orbital but the HOMO correspond to a pi ligand orbital. The full interaction diagram could be intresting. The authors could have represented the densities of spin and thus to check the delocalization of the single electron. This is an important data for applications in magnetism. 

"It should be specially noted that any mention of manganese complexes having I, II and III formulas, in the special literature devoted to (NNNN)‐donor atomic macrocyclic ligands – porphyrins, porphyrazines and their derivatives [7‐13], or anywhere else, was not found by us." Maybe the authors could indicate the non oxo complexes Mn(III) phthalocyanine and Mn(III) porphyrazines decribied in https://doi.org/10.1016/S0020-1693(01)00365-6

Since the +VI is debatable, it could be interesting to compare the data with parent complexes.  

Small typos: 

- First scheme (drawings of I, II, and III) cut a sentence in two. 

- I do not understand this sentence: "To calculate the parameters of molecular structures with a multiplicity greater than 2, we used an unrestricted method. (UB3PW91, UOPBE)." For any multiplicity above 1 the unrestricted method is used by default.  

- Both Sections 3 and 4 are named  Results and Discussion. 

- In supplementary materials some energy units are written in russian (probably kcal/mol)

Cordially, 

Author Response

This paper  presents a  theoretical study comparing  two different  functionals on  a series  of  hypothetical manganese  complexes.  The  methodology  employed  by  the  authors  is  known  to  be reliable  for  metal complexes in general (B3PW91) and especially for spin states energies with the OPBE functional. The chosen basis set is large and reliable. The study details the optimized geometry, frontier molecular orbitals and charges for the compounds that converged. These results are interesting and deserve to be published. However, I think that the journal Materials is not suitable for the study as it is currently presented because the main property discussed here is the feasibility while we would have liked a more important development on other properties such as magnetism or optical characteristics.
In connection with this remark of our esteemed Reviewer 2, we would like to inform you that the paper was presented in the Special Issue "Density Functional Theory Application on Chemical Calculation", in which, according to his intention, articles devoted to the theory and practice of the DFT method for calculating a wide  variety  of  characteristics  of  substances  and  materials  based  on  them,  and  not  just  optical and/or magnetic properties. We, as the authors of this paper, are chemists, and the main direction of our research is, first of all, obtaining data on the spatial structure of the objects we study, which is mainly the subject of this paper. In this regard, we cannot agree with the opinion of our esteemed Reviewer that “the journal Materials is not suitable for the study as it is currently presented because the main property discussed here is the feasibility while we would have liked a more important development on other properties such as magnetism  or  optical  characteristics”.  Moreover,  a  similar  article  with  our  authorship  was  previously published  in  this  journal  (see  Materials  2020,  13,  3162;  doi:10.3390/ma13143162)  and  none  of  the Reviewers or Editors of this article expressed doubts about the relevance of its subject and / or content on the subject of Materials Journal.

Also, I find that the display of the results is a bit inverted. For me, the most important result is the electronic structure. The geometry is a consequence of the electronic structure. The differences in distances (MnO) and angles (OMnO) probably are the results of a different electronic state.
Although we agree in principle with this remark of our esteemed Reviewer, nevertheless, we would like to emphasize once again that for us, as chemists, the greatest interest in this study was precisely the spatial structure of the objects which we considered. We believe that this position of ours has every right to exist.

I also find that the discussion on the redox state is dismissed too quickly. Yes, the definition is based on formal charges based on a simplistic structural formula does not preclude the use of the resources now provided by quantum chemical calculations. Thus the analysis of natural charges (in SI) show a number of 3d electrons quite different to the imagined 3d1 configuration. The metal 
ligand sigma and pi interactions seem to be interesting. The FMO representation does not always correspond to the unpaired electron(s). Like for [Mn(P)(O)2] where the LUMO do correspond to empty an t2g orbital but the HOMO correspond to a pi ligand orbital. The full interaction diagram could be intresting. The authors could have represented the densities of spin and thus to check the  delocalization of the single electron. This is an important data for applications in magnetism.
Fulfilling the wish of our esteemed Reviewer, we have included in the revised text of the paper pictures showing the distribution of spin density in the complexes we are considering.

"It should be specially noted that any mention of manganese complexes having I, II and III formulas, in the special literature devoted to (NNNN)‐donor atomic macrocyclic ligands – porphyrins, porphyrazines and their derivatives [7‐13], or anywhere else, was not found by us." Maybe the authors could indicate the non oxo       complexes       Mn(III)       phthalocyanine       and       Mn(III)       porphyrazines       decribied       in  https://doi.org/10.1016/S0020-1693(01)00365-6.
This wish of our esteemed Reviewer seems to us quite fair, and we have added a link to the article indicated by him in the list of cited literature (see link [10] in the References).

Since the +VI is debatable, it could be interesting to compare the data with parent complexes.
It remains unclear to us what kind of “parent complexes” our esteemed Reviewer had in mind. It is possible that these are manganese homoligand complexes with porphyrazine [Mn(P)] and phthalocyanine [Mn(Pc)] (H2P and H2Pc are porphyrazine and phthalocyanine, respectively). At present, we have calculation data for those variants of the DFT method that were used in our paper, only for the second of these complexes, and if our esteemed Reviewer meant it, then we are ready to immediately present them and include them either in the text of the paper, or in Supplementary Materials, depending on the wishes of the Reviewer. We do not yet have calculated data on [MnP] using these methods, but we can implement them if our esteemed Reviewer considers it necessary, but this will take
some time (according to our estimates, about a month). In this regard, we present a table with structural data for the [MnPc] complex obtained by DFT B3PW91/TZVP DFT OPBE/TZVP methods, for the same molecular sructure parametrs os indicated in Table1 of the revised text of this paper.

Small typos:
- First scheme (drawings of I, II, and III) cut a sentence in two.
We agree with this remark of our Reviewer; the bifurcation of the sentence indicated by him in the revised text of the paper has been eliminated.

- I do not understand this sentence: "To calculate the parameters of molecular structures with a multiplicity greater than 2, we used an unrestricted method. (UB3PW91, UOPBE)." For any multiplicity above 1 the unrestricted method is used by default.
We agree also with this remark of our esteemed Reviewer, and we have made the appropriate correction of the phrase indicated by him (especially since Reviewer 1 also pointed out the need for this in his review).

- Both Sections 3 and 4 are named Results and Discussion.
This error occurred due to our inattention, the corresponding correction in the revised text of the paper has been made.

- In supplementary materials some energy units are written in russian (probably kcal/mol).
This error also occurred due to our carelessness, the corresponding correction in Supplementary Materials is made. The 2 units of energy indicated by our esteemed Reviewer are kJ/mol. (A revised version of the Supplementary Materials is provided along with a revised version of the paper).

All changes and / or additions to the text of the paper, made by us in accordance with the comments of Reviewer 2, are highlighted in the text with a blue fill. One of these remarks, which coincides  in substance with the similar remark of Reviewer 2, is highlighted in green.

On behalf of authors,
Dr. D.V. Chachkov
